*Extended Abstract Track*

# Sheaf Attention Networks

**Federico Barbero**                                FEDERICO.BARBERO@CS.OX.AC.UK
*University of Oxford, Department of Computer Science*

**Cristian Bodnar**                                CB2015@CAM.AC.UK
*University of Cambridge, Department of Computer Science & Technology*

**Haitz Sáez-de-Ocáriz-Borde**                    HAITZ@OXFORDROBOTICS.INSTITUTE
*University of Oxford, Oxford Robotics Institute*

**Pietro Liò**                                    PL219@CAM.AC.UK
*University of Cambridge, Department of Computer Science & Technology*

**Editors:** Sophia Sanborn, Christian Shewmake, Simone Azeglio, Arianna Di Bernardo, Nina Miolane

## Abstract

Attention has become a central inductive bias for deep learning models irrespective of domain. However, increasing theoretical and empirical evidence suggests that Graph Attention Networks (GATs) suffer from the same pathological issues affecting many other Graph Neural Networks (GNNs). First, GAT's features tend to become progressively smoother as more layers are stacked, and second, the model performs poorly in heterophilic graphs. Sheaf Neural Networks (SNNs), a new class of models inspired by algebraic topology and geometry, have shown much promise in tackling these two issues. Building upon the recent success of SNNs and the wide adoption of attention-based architectures, we propose Sheaf Attention Networks (SheafANs). By making use of a novel and more expressive attention mechanism equipped with geometric inductive biases, we show that this type of construction generalizes popular attention-based GNN models to cellular sheaves. We demonstrate that these models help tackle the oversmoothing and heterophily problems and show that, in practice, SheafANs consistently outperform GAT on synthetic and real-world benchmarks.

**Keywords:** Graph Neural Networks, Graph Attention Networks, Sheaf Neural Networks

## 1. Introduction

Graph Neural Networks (GNNs) are Neural Networks which operate over graphs. Their main competitive advantage being that they are able to exploit the graph's local neighbourhood structure in the computations of the latent feature vectors. Two problematic phenomena are associated to many GNN architectures: oversmoothing and the heterophily problem. The former refers to the fact that building very deep neural networks often leads to poor performance. The latter instead is related to the fact that GNNs are commonly built with the inductive bias that neighbouring nodes are likely to be similar. One affected and popular GNN model is the Graph Attention Network (GAT).

It has been shown that the heterophily and oversmoothing problems are, from a topological perspective, intimately connected (Bodnar et al., 2022). GNNs which operate over cellular sheaves, Sheaf Neural Networks (SNNs) (Hansen and Gebhart, 2020; Bodnar et al., 2022), offer a principled way of mitigating the aforementioned issues. In this work, motivated by the great success of attention mechanisms, we propose the Sheaf Attention Network (SheafAN), a generalization of the popular Graph Attention Network to cellular sheaves. We find that SheafAN is able mitigate the issues of oversmoothing and heterophily in GAT by leveraging the more complex geometry present in the sheaf.

Extended Abstract Track

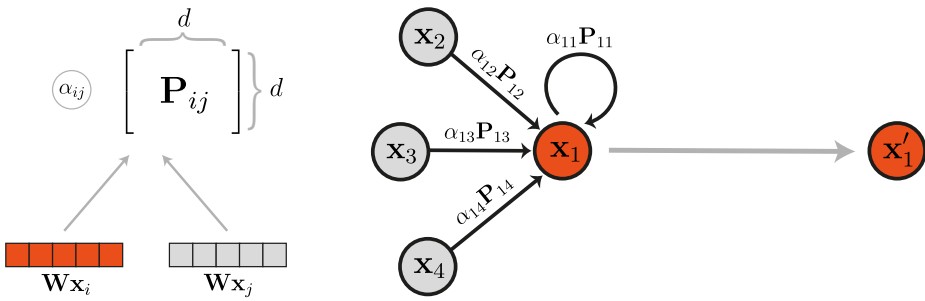

Figure 1: **Left:** Besides the usual attention coefficient $\alpha_{ij}$, the sheaf-based attention mechanism in SheafAN also learns a $d \times d$ "transport matrix" $\mathbf{P}_{ij}$ (typically orthogonal) describing how the vectors in the stalk of node $j$ should be moved to the stalk of node $i$. **Right:** A SheafAN layer updates the features at node $i$ by transporting all the vectors from the neighbouring stalks and aggregating them proportionally to the attention coefficients.

## 2. Background

**Definition 1** *A cellular sheaf $(G, \mathcal{F})$ on an undirected graph $G = (V, E)$ consists of: (1) a vector space $\mathcal{F}(v)$ for each $v \in V$, (2) a vector space $\mathcal{F}(e)$ for each $e \in E$ and, (3) a linear map $\mathcal{F}_{v \trianglelefteq e} : \mathcal{F}(v) \to \mathcal{F}(e)$ for each incident node-edge pair $v \trianglelefteq e$.*

The vector spaces of the node and edges are called *stalks*, while the linear maps are called *restriction maps*. The sheaf $\mathcal{F}$ "bundles" the stalks and restriction maps together. The underlying graph specifies the nodes and edges, while the sheaf specifies a network of linear transformations over the nodes and edges. We define the space of 0-cochains $C^0(G, \mathcal{F})$ as the direct sum over the node stalks $C^0(G, \mathcal{F}) := \bigoplus_{v \in V} \mathcal{F}(v)$. Similarly, the space of 1-cochains $C^1(G, \mathcal{F})$ as the direct sum over the edge stalks $C^1(G, \mathcal{F}) := \bigoplus_{e \in E} \mathcal{F}(e)$. It is then natural to construct a co-boundary operator $\delta : C^0(G, \mathcal{F}) \to C^1(G, \mathcal{F})$ as $\delta(\mathbf{x})_e = \mathcal{F}_{v \trianglelefteq e} \mathbf{x}_v - \mathcal{F}_{u \trianglelefteq e} \mathbf{x}_u$. Through the co-boundary operator, we can construct a sheaf Laplacian. The sheaf Laplacian of a sheaf is a map $\mathbf{L}_\mathcal{F} : C^0(G, \mathcal{F}) \to C^0(G, \mathcal{F})$ defined as $\mathbf{L}_\mathcal{F} = \delta^\top \delta$. The *normalized sheaf Laplacian* $\Delta_\mathcal{F}$ is defined as $\Delta_\mathcal{F} = \mathbf{D}^{-\frac{1}{2}} \mathbf{L}_\mathcal{F} \mathbf{D}^{-\frac{1}{2}}$ where $\mathbf{D}$ is the block-diagonal of $\mathbf{L}_\mathcal{F}$. The sheaf Laplacian is a symmetric positive semi-definite (by construction) block matrix. The diagonal blocks are $\mathbf{L}_{\mathcal{F}_{v,v}} = \sum_{v \trianglelefteq e} \mathcal{F}_{v \trianglelefteq e}^\top \mathcal{F}_{v \trianglelefteq e}$, while the off-diagonal blocks are $\mathbf{L}_{\mathcal{F}_{v,u}} = -\mathcal{F}_{v \trianglelefteq e}^\top \mathcal{F}_{u \trianglelefteq e}$.

With this machinery, one is able to build a Sheaf Neural Network layer (Hansen and Gebhart, 2020; Bodnar et al., 2022), a GNN layer which operates over cellular sheaves. This arises from the Euler discretization of a sheaf heat diffusion PDE (Hansen and Ghrist, 2021). Consider a graph $G$ with $n$ nodes and $d$-dimensional node feature vectors $\mathbf{x}_v \in \mathcal{F}(v)$. Their concatenation $\mathbf{x}$ is a 0-cochain such that $\mathbf{x} \in C^0(G, \mathcal{F})$. Allowing for $f$ feature channels, we can construct a $(nd) \times f$ feature matrix $\mathbf{X}$. The Neural Sheaf Diffusion (Bodnar et al., 2022) model takes the form:

$$\mathbf{X}_{t+1} = \mathbf{X}_t - \sigma \left( \Delta_{\mathcal{F}(t)} \left( \mathbf{I}_n \otimes \mathbf{W}_1^t \right) \mathbf{X}_t \mathbf{W}_2^t \right), \tag{1}$$

where $\sigma$ is a non-linearity, $\mathbf{I}_n$ is a $n \times n$ identity matrix, and $\mathbf{W}_1^t$ and $\mathbf{W}_2^t$ are learnable weight matrices. The sheaf Laplacian at time $t$, $\Delta_{\mathcal{F}(t)}$, is typically learnt through gradient-based approaches by learning the individual restriction maps with a MLP $\Phi$, i.e. $\mathcal{F}_{v \trianglelefteq e := (v,u)} = \Phi(\mathbf{x}_v, \mathbf{x}_u)$. More recently, Barbero et al. (2022) have shown that the sheaf can also be learnt using tools inspired by Riemannian geometry.

## 3. Sheaf Attention Networks

The sheaf-attention mechanism we employ relies on an attention matrix $\Lambda$, which is a $n \times n$ row-stochastic matrix. Each entry $\Lambda_{ij}$ is computed with an attention function $a(\mathbf{x}_i, \mathbf{x}_j)$:

$$\Lambda_{ij} = a(\mathbf{x}_i, \mathbf{x}_j) = \frac{\exp\left(\text{LeakyRelu}\left(\mathbf{a}[\mathbf{W}\mathbf{x}_i || \mathbf{W}\mathbf{x}_j]\right)\right)}{\sum_{k \in \mathcal{N}_i} \exp\left(\text{LeakyRelu}\left(\mathbf{a}[\mathbf{W}\mathbf{x}_i || \mathbf{W}\mathbf{x}_k]\right)\right)}, \tag{2}$$

where $\mathbf{a}$ and $\mathbf{W}$ are learnable weights of appropriate dimension, and $||$ denotes vector concatenation. We employ multi-head attention which empirically tends to outperform single-head mechanisms (Velickovic et al., 2017; Vaswani et al., 2017). Note that this is the same attention mechanism utilised in GAT. To generalise $\Lambda$ to $d$-dimensional sheaves, we construct the sheaf-attention matrix $\hat{\Lambda}$ such that $\hat{\Lambda} = \Lambda \otimes \mathbf{1}_d$ where $\mathbf{1}_d$ is a $d \times d$-dimensional matrix with every element 1 and $\otimes$ is the Kronecker product. We then consider the following attentive sheaf diffusion PDE, which evolves the node features $\mathbf{X}$ over time:

$$\mathbf{X}(0) = \mathbf{X}, \quad \frac{\partial}{\partial t}\mathbf{X}(t) = \left(\hat{\Lambda}(\mathbf{X}) \odot \hat{\mathbf{A}}_{\mathcal{F}} - \mathbf{I}\right)\mathbf{X}(t). \tag{3}$$

Here $\odot$ indicates element-wise multiplication, and $\hat{\mathbf{A}}_{\mathcal{F}}$ is the sheaf adjacency matrix with added self-loops. That is $\hat{\mathbf{A}}_{\mathcal{F}}(i,j) = \mathcal{F}_{i \trianglelefteq e}^\top \mathcal{F}_{j \trianglelefteq e} =: \mathbf{P}_{ij}$. The Euler discretisation with unit time-step ($\tau = 1$) of Equation 3 has the form:

$$\mathbf{X}(t+1) = \left(\hat{\Lambda}(\mathbf{X}) \odot \hat{\mathbf{A}}_{\mathcal{F}}\right)\mathbf{X}(t). \tag{4}$$

We can equip Equation 4 with weight matrices $\mathbf{W}_t^1 \in \mathbb{R}^{d \times d}$ and $\mathbf{W}_t^2 \in \mathbb{R}^{f_t \times f_{t+1}}$ and a non-linearity $\sigma$ to derive our new *Sheaf Attention Network* (SheafAN) layer (see Figure 1):

$$\mathbf{X}_{t+1} = \sigma\left(\left(\hat{\Lambda}(\mathbf{X}_t) \odot \mathbf{A}_{\mathcal{F}}\right)\left(\mathbf{I}_n \otimes \mathbf{W}_t^1\right)\mathbf{X}_t \mathbf{W}_t^2\right), \tag{5}$$

where $f_t$ and $f_{t+1}$ are the input and output channels sizes respectively, and $\hat{\Lambda}(\mathbf{X}_t)$ is the attention matrix for layer $t$. It is natural to call this a Sheaf Attention Network, since the original GAT model is a particular instantiation of the aforementioned model for which restriction maps are 1-dimensional ($d = 1$) and equal to 1. We also consider a different parametrization of Equation 4:

$$\mathbf{X}_{t+1} = \mathbf{X}_t + \sigma\left(\left(\hat{\Lambda}(\mathbf{X}_t) \odot \mathbf{A}_{\mathcal{F}} - \mathbf{I}\right)\left(\mathbf{I}_n \otimes \mathbf{W}_t^1\right)\mathbf{X}_t \mathbf{W}_t^2\right), \tag{6}$$

which we call Res-SheafAN. The Res-SheafAN model is different to the SheafAN model as it uses a type of residual parametrization, where the feature updates are of the form

Extended Abstract Track

$\mathbf{h}_{t+1} = \mathbf{h}_t + f(\mathbf{h}_t, \theta_t)$. This is motivated from the spectral perspective of being able to act as both a high-pass and low-pass filter (Di Giovanni et al., 2022). In particular, without a residual connection the model may be dominated by the low frequency signals over the graph as more layers are added, contributing to oversmoothing.

For our purposes, we use orthogonal restriction maps, i.e. $\mathcal{F}_{v \trianglelefteq e} \in O(d)$, effectively learning an $O(d)$-vector bundle. Since orthogonal transformations are norm-preserving, the attention coefficients play the complementary role of adjusting the strength or magnitude of the incoming messages. Besides, $O(d)$ restriction maps are advantageous because they have $d(d-1)/2$ free parameters compared to the $d^2$ free parameters in general linear maps, which can be understood as a form of regularization. On top of this, they have also been shown to be theoretically advantageous as they are able to more efficiently separate nodes using the same stalk width $d$ when compared to, for instance, diagonal linear maps (Bodnar et al., 2022).

Encoding signed messages (using rotation angles of orthogonal maps for example) is crucial in settings of low homophily (Yan et al., 2021). From an opinion dynamics perspective (Hansen and Ghrist, 2021), a negatively signed message may be thought of as modelling a node's opinion which contradicts another node's opinion. As one can imagine, this behaviour is related to low homophily, where two nodes are more likely nodes to "disagree" on their class. We therefore expect this higher-dimensional disagreement in the form of higher-dimensional orthogonal restriction maps to be very advantageous in heterophilic settings.

Table 1: Accuracy ± stdev for various node classification datasets. "OOM" stands for out of memory, whilst "INS" stands for numerical instability. The top three models for each dataset and layer count are coloured by **First**, **Second** and **Third**.

| Layers | 2 | 4 | 8 | 16 | 32 | 64 | Best | 2 | 4 | 8 | 16 | 32 | 64 | Best |
|---|---|---|---|---|---|---|---|---|---|---|---|---|---|---|
| | Cora (h=0.81) | | | | | | | Citeseer (h=0.74) | | | | | | |
| SheafAN (ours) | 86.90±1.31 | 86.84±0.97 | 86.68±1.13 | 86.54±0.89 | 86.62±1.39 | 86.26±1.09 | 2 | 76.27±1.76 | 76.30±1.80 | 76.62±1.70 | 76.18±1.47 | 76.07±2.18 | 75.99±1.84 | 8 |
| Res-SheafAN (ours) | 86.98±1.07 | 87.08±1.26 | 86.80±1.15 | 86.84±1.24 | 86.56±0.75 | 86.76±1.49 | 4 | 76.99±1.74 | 76.86±1.71 | 76.61±1.51 | 76.69±1.56 | 76.22±1.47 | 76.44±1.30 | 2 |
| GGCN | 87.00±1.15 | 87.48±1.32 | 87.63±1.33 | 87.51±1.19 | 87.95±1.05 | 87.28±1.41 | 32 | 76.83±1.82 | 76.77±1.48 | 76.91±1.56 | 76.88±1.56 | 76.97±1.52 | 76.65±1.38 | 10 |
| GPRGNN | 87.93±1.11 | 87.95±1.18 | 87.87±1.41 | 87.26±1.51 | 87.18±1.29 | 87.32±1.21 | 4 | 77.13±1.67 | 77.05±1.43 | 77.09±1.62 | 76.00±1.64 | 74.97±1.47 | 74.41±1.65 | 2 |
| H2GCN | 87.87±1.20 | 86.10±1.51 | 86.18±2.10 | OOM | OOM | OOM | 2 | 76.90±1.80 | 76.09±1.54 | 74.10±1.83 | OOM | OOM | OOM | 1 |
| GCNII | 85.35±1.56 | 85.35±1.48 | 86.38±0.98 | 87.12±1.11 | 87.95±1.23 | 88.37±1.25 | 64 | 75.42±1.78 | 75.29±1.90 | 76.00±1.66 | 76.96±1.38 | 77.33±1.48 | 77.18±1.47 | 32 |
| PairNorm | 85.79±1.01 | 85.07±0.91 | 84.65±1.09 | 82.21±2.84 | 60.32±8.28 | 44.39±5.60 | 2 | 73.59±1.47 | 72.62±1.97 | 72.32±1.58 | 59.71±15.97 | 27.21±10.95 | 23.82±6.64 | 2 |
| Geom-GCN | 85.35±1.57 | 21.01±2.61 | 13.98±1.48 | 13.98±1.48 | 13.98±1.48 | 13.98±1.48 | 2 | 78.02±1.15 | 23.01±1.95 | 7.23±0.87 | 7.23±0.87 | 7.23±0.87 | 7.23±0.87 | 2 |
| GCN | 86.98±1.27 | 83.24±1.56 | 31.03±3.08 | 31.05±2.36 | 30.76±3.43 | 31.89±2.08 | 2 | 76.50±1.36 | 64.33±8.27 | 24.18±1.71 | 23.07±2.95 | 25.3±1.77 | 24.73±1.66 | 2 |
| GAT | 87.30±1.10 | 86.50±1.20 | 84.97±1.24 | INS | INS | INS | 2 | 76.55±1.23 | 75.33±1.39 | 66.57±5.08 | INS | INS | INS | 2 |
| | Cornell (h=0.3) | | | | | | | Chameleon (h=0.23) | | | | | | |
| SheafAN (ours) | 82.70±6.64 | 84.59±4.69 | 85.68±4.53 | 84.32±6.82 | 83.51±7.20 | 78.65±10.29 | 2 | 65.88±2.10 | 65.99±2.24 | 68.16±2.18 | 68.62±2.81 | 67.61±2.80 | OOM | 16 |
| Res-SheafAN (ours) | 84.86±6.07 | 84.32±5.10 | 84.86±5.95 | 84.59±6.51 | 83.24±3.97 | 82.43±8.90 | 8 | 65.35±1.26 | 67.11±1.66 | 66.69±2.30 | 67.39±1.84 | 66.91±1.61 | OOM | 16 |
| GGCN | 83.78±6.73 | 83.78±6.16 | 84.86±5.69 | 83.78±6.73 | 83.78±6.51 | 84.32±5.90 | 6 | 70.77±1.42 | 69.58±2.66 | 70.33±1.70 | 70.44±1.82 | 70.29±1.62 | 70.20±1.95 | 5 |
| GPRGNN | 76.76±8.22 | 77.57±7.46 | 80.27±8.11 | 78.38±6.04 | 74.59±7.66 | 70.00±5.73 | 8 | 46.58±1.771 | 45.72±3.45 | 41.16±5.79 | 39.58±7.85 | 35.42±8.52 | 36.38±2.40 | 2 |
| H2GCN | 81.89±5.98 | 82.70±6.27 | 80.27±6.63 | OOM | OOM | OOM | 1 | 59.06±1.85 | 60.11±2.15 | OOM | OOM | OOM | OOM | 4 |
| GCNII* | 67.57±11.34 | 64.59±9.63 | 73.24±5.91 | 77.84±3.97 | 75.41±5.47 | 73.78±4.37 | 16 | 61.07±4.10 | 63.86±3.04 | 62.89±1.18 | 60.20±2.10 | 56.97±1.81 | 55.99±2.27 | 4 |
| PairNorm | 50.27±7.17 | 53.51±8.00 | 58.38±5.01 | 58.38±3.01 | 58.92±3.15 | 58.92±3.15 | 32 | 62.74±2.82 | 59.01±2.80 | 54.12±2.24 | 46.38±2.23 | 46.78±2.26 | 46.27±3.24 | 2 |
| Geom-GCN* | 60.54±3.67 | 23.78±11.64 | 12.97±2.91 | 12.97±2.91 | 12.97±2.91 | 12.97±2.91 | 2 | 60.00±2.81 | 19.17±1.66 | 19.58±1.73 | 19.58±1.73 | 19.58±1.73 | 19.58±1.73 | 2 |
| GCN | 60.54±5.30 | 59.19±3.30 | 58.92±3.15 | 58.92±3.15 | 58.92±3.15 | 58.92±3.15 | 2 | 64.82±2.24 | 53.11±4.44 | 35.15±3.14 | 35.39±3.23 | 35.20±3.25 | 35.50±3.08 | 2 |
| GAT | 61.89±5.05 | 58.38±4.05 | 58.38±3.86 | INS | INS | INS | 2 | 60.26±2.50 | 48.71±2.96 | 35.09±3.55 | INS | INS | INS | 2 |

**Evaluation** We evaluate the performance of SheafAN and Res-SheafAN in different settings of heterophily and with varying number of layers. We follow the analysis structure proposed by Yan et al. (2021) by evaluating our sheaf attention models on Cora, Citeseer, Cornell and Chameleon with layers increasing in powers of 2 from 2 to 64. The accuracy and standard deviation is computed with respect to 10 fixed splits provided by (Pei et al., 2020), with each split containing 48%/32%/20% of nodes for training/calibration/testing

respectively. We run our experiments on a machine equipped with an NVIDIA TITAN X GPU (12 GB) and an Intel(R) Core(TM) i7-6700 CPU @ 3.40 GhZ.

We consider models of different classes. 1) Classical GNN models: GCN (Kipf and Welling, 2016), and GAT (Velickovic et al., 2017). 2) Models designed to address over-smoothing: GCNII (Chen et al., 2020), and PairNorm (Zhao and Akoglu, 2019). 3) Models designed for heterophilic settings: GGCN (Yan et al., 2021), Geom-GCN (Pei et al., 2020), H2GCN (Zhu et al., 2020), and GPRGNN (Chien et al., 2020). Table 1 shows the over-smoothing results. We see how models which are not designed for oversmoothing exhibit various issues, ranging from accuracy dropping rapidly (GCN and Geom-GCN), to memory issues (H2GCN), to numerical instability (GAT).

We find that our sheaf attention mechanism models perform very competitively when compared to state of the art techniques such as GCNII. Furthermore, SheafAN consistenly outperforms GAT. We attribute this to the more complex geometry of the underlying sheaf. The model is able to leverage the additional stalk width to tamper the effect of oversmoothing, in accordance with the analysis of Bodnar et al. (2022). SheafAN performs particularly well on the Cornell and Chameleon datasets, when compared to the other models, which are particularly heterophilic datasets. While other models are generally designed to combat either oversmoothing or heterophily, SheafAN seems to be particularly well-equipped for both at once.

**Conclusion** We proposed SheafAN, a generalization of GAT, which makes use of an additional cellular sheaf structure on top of the graph. The resulting form of sheaf-based attention demonstrates competitive performance with state-of-the-art models on tasks related to oversmoothing and heterophily. We hope the present work will encourage further integration between sheaves and the attention mechanisms often employed in deep learning models.

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

# Extended Abstract Track

Francesco Di Giovanni, James Rowbottom, Benjamin P Chamberlain, Thomas Markovich, and Michael M Bronstein. Graph neural networks as gradient flows. *arXiv preprint arXiv:2206.10991*, 2022.

Jakob Hansen and Thomas Gebhart. Sheaf neural networks. *arXiv preprint arXiv:2012.06333*, 2020.

Jakob Hansen and Robert Ghrist. Opinion dynamics on discourse sheaves. *SIAM Journal on Applied Mathematics*, 81(5):2033–2060, 2021.

Thomas N Kipf and Max Welling. Semi-supervised classification with graph convolutional networks. *arXiv preprint arXiv:1609.02907*, 2016.

Hongbin Pei, Bingzhe Wei, Kevin Chen-Chuan Chang, Yu Lei, and Bo Yang. Geom-gcn: Geometric graph convolutional networks. *arXiv preprint arXiv:2002.05287*, 2020.

Ashish Vaswani, Noam Shazeer, Niki Parmar, Jakob Uszkoreit, Llion Jones, Aidan N Gomez, Łukasz Kaiser, and Illia Polosukhin. Attention is all you need. *Advances in neural information processing systems*, 30, 2017.

Petar Velickovic, Guillem Cucurull, Arantxa Casanova, Adriana Romero, Pietro Lio, and Yoshua Bengio. Graph attention networks. *stat*, 1050:20, 2017.

Yujun Yan, Milad Hashemi, Kevin Swersky, Yaoqing Yang, and Danai Koutra. Two sides of the same coin: Heterophily and oversmoothing in graph convolutional neural networks. *arXiv preprint arXiv:2102.06462*, 2021.

Lingxiao Zhao and Leman Akoglu. Pairnorm: Tackling oversmoothing in gnns. *arXiv preprint arXiv:1909.12223*, 2019.

Jiong Zhu, Yujun Yan, Lingxiao Zhao, Mark Heimann, Leman Akoglu, and Danai Koutra. Generalizing graph neural networks beyond homophily. *arXiv preprint arXiv:2006.11468*, 2020.

