# OpenReview forum: "Sheaf Attention Networks"
_NeurIPS.cc/2022/Workshop/NeurReps — NeurReps 2022 Oral_

### Official Review · Reviewer_L4nh · 2022-10-09

**Confidence:** 4
**Soundness:** 3
**Presentation:** 4
**Contribution:** 3
**Overall Rating:** 7

**Summary:**

The authors propose a novel attention-based architecture derived from Sheaf Neural Networks. The model tackles such problems of graph attention networks as oversmoothing and poor performance on heterophilic graphs. The authors demonstrate that the proposed model outperforms its non-sheaf attention counterpart (GAT) on both synthetic and real-world data regarding classification accuracy while being resistant to oversmoothing.


**Questions:**

- how many runs did you have for each experimental setup (Table 1)?

**Limitations:**

The authors did not provide any limitations. Perhaps, computational time and memory consumption would be helpful.

**Recommended Decision:**

3: Accept

**Relevance:**

4: Highly relevant

**Strengths And Weaknesses:**

**Strengths**
- incredibly well-written.
- the theory is well delivered and easily digestible, which is appreciated as sheaf theory is (yet) not widespread in the community.
- the model is compared to an extensive set of baseline approaches on multiple datasets.
- the results are promising.

**Weaknesses**
- in Table 1, the values of standard deviation are pretty high everywhere. Thus, the current number of experiments might not be sufficient to draw conclusions as differences are not statistically significant in some cases. More runs for each experiment are desirable.
- the number of parameters and inference time are not provided.
- hardware information is not given. It would be helpful to put "out-of-memory" in context (Table 1).


**Submission Track:**

Extended Abstract (4 Page)

---

### Official Review · Reviewer_LLkx · 2022-10-15
**Review of Sheaf attention networks**

**Confidence:** 3
**Soundness:** 3
**Presentation:** 4
**Contribution:** 3
**Overall Rating:** 6

**Summary:**

The manuscript introduces a novel approach for graph learning which combines sheaf diffusion on graphs and the attention mechanism. The latter is extended to be compatible with the sheaf
Two different formulations, a vanilla and a residual one,  of the attention mechanisms are proposed. The main purpose of the approach is to propose a graph learnin algorithm which is able to deal with oversmoothing and heterophily of graphs, which typically affects negatively the performance of most classical architectures on graphs. The approach is validated experimentally on different graph benchmars, including datasets which present high heterophily of the nodes features.






**Questions:**

-From an experimental perspective (referring to table 1) how the performance of the proposed method compares to the vanilla sheaf neural networks
-Referring to table 1, why does the performance increase the number of layers not increase monotonically with respect to the accuracy? More generally it doesn't seem that the depth of the network is correlated with the accuracy score, even though the sheaf approach should counteract the effect of over smoothing. (as observed in GGCN (chien et al, 2020))



**Limitations:**

The main limitation of the work at its current state is that it is difficult to assess the contribution of the approach. In this regard the most straightforward way to tackle this, would be to include experimental results on the same datasets with sheaf neural networks Also of interest would be to characterize if the attention mechanism on top of the sheaf framework is able to enrich the expressivity of the network, allowing to compute a larger class of functions.


**Recommended Decision:**

3: Accept

**Relevance:**

3: Solid fit

**Strengths And Weaknesses:**

*Originality*

The method proposed is original as it combines two previous approaches (sheaf neural networks and graph attention networks) to obtain a novel approach to better tackle oversmoothing and graph heterophily.

*Quality*

To the best of my judgement, the paper is technically sound. The experimental section is quite rich although it misses comparisons with vanilla (i.e. without attention) sheaf neural networks (eg. Bodnar 2022). This makes more difficult to assess the contribution of the attention module on top of the existing neural sheaf diffusion, which is the core contribution of the work.

*Clarity*

The paper is clear and well written.

*Significance*
The approach proposed is of interest to tackle the problem of oversmoothing and  : to consolidate the contribution of the paper it would be interesting further to add experimental evidence of the benifits of the attention mechanism in the sheaf formalism and / or theorethical analysis (see limitations)  concerning the expressive power of the network.

**Submission Track:**

Extended Abstract (4 Page)

---

### Official Review · Reviewer_M73N · 2022-10-18
**Recommend to Accept**

**Confidence:** 3
**Soundness:** 3
**Presentation:** 3
**Contribution:** 3
**Overall Rating:** 7

**Summary:**

The authors propose an extension of Sheaf Neural Networks by considering attention weights that are multiplied to the morphisms going from vertices to neighboring vertices and benchmark their method on 4 datasets. GATs are a particular case for constant functors to $\mathbb{R}$.

**Questions:**


Could the authors motivate why they consider only O(n) and not any matrix?

Do the authors see other ways than the increasing layer analysis (Table 1) to asses oversmoothing and heterophily?

**Limitations:**

Addressed (Table 1)

**Recommended Decision:**

3: Accept

**Relevance:**

4: Highly relevant

**Strengths And Weaknesses:**

Originality: The work is a straightforward extension combining GATs and Sheaf Neural Networks but writing it correctly can be tricky; this shows that authors did a very good job at making it clear. The result stays very interesting.

Quality: The equations are clear and sound and previous results Equations (1) and (2) are given clear references Barbero et al. 2022, Velickovic et al., 2017. The link between equation (5) of this paper  and (4) of Velickovic et al., 2017 could be optionally discussed.

Clarity: The paper is very clear and very easy to read. The second sentence of section 3 where $\Lambda$ relates to the attention function $a$ could be a bit confusing in the way it is stated as, if I understand well, $\Lambda$ and $a$ are the same.


Significance : I believe the results to be of interest for the geometric deep learning community


**Submission Track:**

Extended Abstract (4 Page)

---

### Decision · Program_Chairs · 2022-10-21

Accept (Oral)